# An engineered biosensor enables dynamic aspartate measurements in living cells

Kristian Davidsen[1,2], Jonathan S Marvin[3]*, Abhi Aggarwal[3], Timothy A Brown[3], Lucas B Sullivan[1]*

[1]Human Biology Division, Fred Hutchinson Cancer Center, Seattle, United States; [2]Molecular and Cellular Biology Program, University of Washington, Seattle, United States; [3]Howard Hughes Medical Institute (HHMI), Janelia Research Campus, Ashburn, United States

*For correspondence:
marvinj@janelia.hhmi.org (JSM);
lucas@fredhutch.org (LBS)

Competing interest: The authors declare that no competing interests exist.

**Abstract** Intracellular levels of the amino acid aspartate are responsive to changes in metabolism in mammalian cells and can correspondingly alter cell function, highlighting the need for robust tools to measure aspartate abundance. However, comprehensive understanding of aspartate metabolism has been limited by the throughput, cost, and static nature of the mass spectrometry (MS)-based measurements that are typically employed to measure aspartate levels. To address these issues, we have developed a green fluorescent protein (GFP)-based sensor of aspartate (jAspSnFR3), where the fluorescence intensity corresponds to aspartate concentration. As a purified protein, the sensor has a 20-fold increase in fluorescence upon aspartate saturation, with dose-dependent fluorescence changes covering a physiologically relevant aspartate concentration range and no significant off target binding. Expressed in mammalian cell lines, sensor intensity correlated with aspartate levels measured by MS and could resolve temporal changes in intracellular aspartate from genetic, pharmacological, and nutritional manipulations. These data demonstrate the utility of jAspSnFR3 and highlight the opportunities it provides for temporally resolved and high-throughput applications of variables that affect aspartate levels.

## eLife assessment

This **important** study reports jAspSnFR3, a biosensor that enables high spatiotemporal resolution of aspartate levels in living cells. To develop this sensor, the authors used a structurally guided amino acid substitution in a glutamate/aspartate periplasmic binding protein to switch its specificity towards aspartate. The in vitro and in cellulo functional characterization of the biosensor is **convincing**, but evidence of the sensor's effectiveness in detecting small perturbations of aspartate levels and information on its behavior in response to acute aspartate elevations in the cytosol are still lacking.

## Introduction

The primary tool used by metabolism researchers, mass spectrometry (MS) coupled with either gas chromatography (GCMS) or liquid chromatography (LCMS), involves extracting pools of thousands of cells and measuring the liberated metabolites. This approach is powerful but has significant drawbacks; it requires highly specialized equipment, it is expensive, and sample preparation by chemical extractions homogenizes metabolic differences that may occur among different cells in complex samples or across subcellular compartments. Metabolite extraction also consumes precious samples

that might otherwise be desirable to analyze over time or with additional outputs. Development of genetically encoded protein sensors (biosensors) over the past two decades has provided new opportunities to visualize the release, production, and depletion of important signaling molecules and metabolites with subsecond and subcellular resolution (reviewed in *Kostyuk et al., 2019*; *Koveal et al., 2020*). Thus, with the trade-off of only monitoring one metabolite per sensor, biosensors provide a solution to many of the problems inherent to metabolite extraction and MS.

Aspartate is among the most concentrated metabolites in cells (*Park et al., 2016*), yet it is one of only two amino acids that is not predominantly acquired from the environment. While the other, glutamate, is made from glutamine by the enzyme glutaminase, no analogous enzyme exists in humans to convert asparagine to aspartate (*Sullivan et al., 2018*). Instead, aspartate must be synthesized by transamination of the tricarboxylic acid cycle metabolite oxaloacetate by the cytosolic enzyme GOT1 or the mitochondrial enzyme GOT2. Notably, aspartate synthesis can occur from multiple metabolic sources via complex metabolic reactions occurring in both the cytosol and mitochondria, rendering aspartate levels at the whole cell and subcellular levels dependent on multiple metabolic variables. For example, impairments to mitochondrial respiration can deplete aspartate levels and aspartate restoration can reestablish proliferation in cells with defective mitochondria (*Sullivan et al., 2015*; *Birsoy et al., 2015*; *Cardaci et al., 2015*; *Hart et al., 2023*). Alterations to aspartate levels are associated with modifications to cell function in multiple biological processes, including stem cells (*Tournaire et al., 2022*; *Arnold et al., 2022*), immune cells (*Bailis et al., 2019*), endothelial cells (*Diebold et al., 2019*), and cancer (*Helenius et al., 2021*). In addition, genetic methods to elevate intracellular aspartate can impact biology in vivo, increasing tumor growth (*Sullivan et al., 2018*; *Garcia-Bermudez et al., 2018*) and improving hematopoietic function (*Qi et al., 2021*). Therefore, our understanding of metabolism in multiple biological systems could be improved with the availability of an aspartate biosensor.

We have previously developed a biosensor for glutamate (iGluSnFR) using the *E. coli* glutamate/aspartate-binding domain (GltI) linked to circularly permutated GFP (*Marvin et al., 2013*), and subsequently optimized it by modulating its affinity, kinetics, color, and total fluorescence change (SF-iGluSnFR and iGluSnFR3) (*Marvin et al., 2018*; *Aggarwal et al., 2023*). Since the GltI domain also binds aspartate, albeit at lower affinity than glutamate (*Hu et al., 2008*), we reasoned that subtle modifications to the ligand-binding site could switch the relative aspartate/glutamate specificity. We achieved this using a small mutagenesis screen on a precursor to iGluSnFR3 (*Supplementary file 1*), guided by the crystal structure of glutamate-bound GltI. The resulting biosensor, jAspSnFR3, was characterized in vitro and in cells with matched LCMS determined aspartate levels, showing that it accurately reports genetic, pharmacological, and nutritional manipulation of intracellular aspartate.

## Results

### Protein engineering

We observed that the glutamate sensor, iGluSnFR, binds both glutamate and aspartate, with higher affinity for the former (*Marvin et al., 2013*). To shift the relative affinities of the two ligands, we evaluated the structure of the binding pocket (*Hu et al., 2008*), and sampled all possible amino acid substitutions of residue S72, which interacts with the side-chain carboxylate of bound glutamate (*Figure 1A*). By expressing mutant sensors in bacteria and measuring the fluorescence of bacterial lysate in response to aspartate and glutamate, we identified S72A and S72P as having switched specificity from glutamate to aspartate. S72T, identified in a faster version of iGluSnFR (*Helassa et al., 2018*), also preferentially binds aspartate over glutamate.

As an improved glutamate sensor (iGluSnFR3) was being developed (*Aggarwal et al., 2023*), we took a variant from that process and queried the effect of S72A, S72T, and S72P on aspartate/glutamate affinity. In bacterial cell lysate, S72P maintained the expected shift to a preference for aspartate when inserted into a iGluSnFR3 precursor, and had a higher fluorescence fold increase ($\Delta F/F$) than either S72A or S72T (*Figure 1—figure supplement 1A*). To further increase specificity of the S72P mutant, we sampled mutations at S27, which also interacts with the carboxylate of bound glutamate. One of those, S27A, had lower affinity for glutamate while mostly maintaining affinity for aspartate (*Figure 1—figure supplement 1A*). We then moved forward with this variant, and since it is built from a precursor of iGluSnFR3, named it Janelia-developed Aspartate-Sensing Fluorescent Reporter

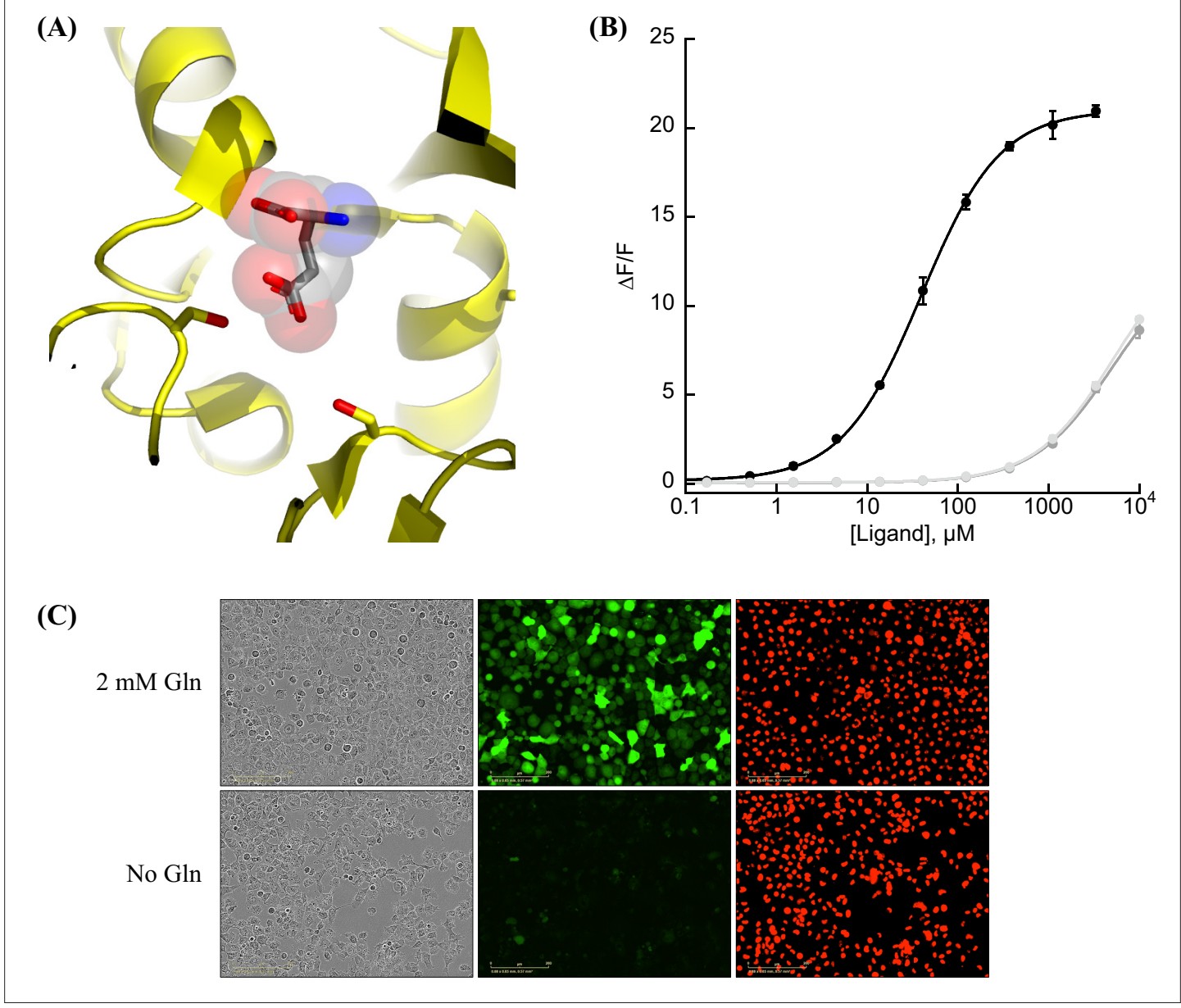

**Figure 1.** Protein engineering and in vitro characterization. (**A**) Structure of the binding pocket of glutamate-bound GltI (2VHA.pdb) with residues S72 (left) and S27 (right) shown as sticks and bound glutamate as sticks inside transparent spheres. (**B**) Fluorescence response of purified jAspSnFR3-mRuby3 when titrated with aspartate (black) or glutamate or asparagine (gray tones). Ex. 485 nm (20 nm bandpass), Em. 535 nm (20 nm bandpass). Error bars are standard deviation (SD) of three technical replicates. (**C**) Live cell imaging in the phase contrast, GFP and red fluorescent protein (RFP) channels of H1299 Nuclear-RFP cells expressing jAspSnFR3 after 24 hr with/without glutamine.

The online version of this article includes the following source data and figure supplement(s) for figure 1:

**Source data 1.** Fluorescence measurements for *Figure 1*.

**Figure supplement 1.** Aspartate specificity and excitation/emission spectra.

**Figure supplement 1—source data 1.** Fluorescence measurements for *Figure 1—figure supplement 1*.

**Figure supplement 2.** Decoy, temperature, and pH sensitivity.

**Figure supplement 2—source data 1.** Fluorescence measurements for *Figure 1—figure supplement 2*.

**Figure supplement 3.** Stopped-flow kinetics.

**Figure supplement 3—source data 1.** Fluorescence measurements and $k_{obs}$ values for *Figure 1—figure supplement 3*.

**Figure supplement 4.** mRuby3 interaction with histidine tag.

**Figure supplement 4—source data 1.** Fluorescence measurements for *Figure 1—figure supplement 4*.

(jAspSnFR3). Since we expected to be using this sensor in cell culture studies, and potentially in vivo, over the course of hours or even days, we added a C-terminal red fluorescence protein, mRuby3, to enable correction for expression. All biochemical characterization is reported with jAspSnFR3-mRuby3. For jAspSnFR3 signal normalization in cells we used a mix of jAspSnFR3-mRuby3 and nuclear localized mRuby2.

Further characterization of the sensor found it is yellow shifted in excitation and emission compared to typical GFP-based sensors, since its chromophore is formed by the triad of GYG and has the T203Y pi-stacking mutations of the Venus yellow fluorescent protein. This yellow shift facilitates observation deeper into tissues using two-photon microscopy, as the sensor has high signal fold change and significant two-photon cross-section at 1040 nm (*Figure 1—figure supplement 1B*). It has a $K_D$ for aspartate of about 50 µM and binds glutamate and asparagine with $K_D$ greater than 5 mM (*Figure 1B*). The sensor also does not appreciably change its green fluorescence in response to other amino acids (*Figure 1—figure supplement 2A*) or to other decoys considered relevant to aspartate metabolism, including pharmacological treatments (*Figure 1—figure supplement 2B*). We performed rapid-mixing stopped-flow fluorescence spectroscopy to determine the aspartate-binding kinetics and measured an on-rate of 0.8 µM$^{-1}$ s$^{-1}$ and an off-rate of 26 s$^{-1}$, resulting in a $K_D$ of 33 µM in agreement with equilibrium measurements (*Figure 1—figure supplement 3*). The off-rate corresponds to a time to equilibrium of 130 ms when defined using five half-lives (*Jarmoskaite et al., 2020*).

Surprisingly, the mRuby3 fluorescence of affinity-purified jAspSnFR3.mRuby3 responds to some amino acids at high millimolar concentrations, indicating a non-specific effect (*Figure 1—figure supplement 4A*). This was determined to be due to an unexpected interaction with the C-terminal histidine tag and could be reproduced with other proteins containing mRuby3 and purified via the same C-terminal histidine tag (*Figure 1—figure supplement 4B, C*). Interestingly, a structurally related, non-amino acid compound, GABA, does not elicit a change in red fluorescence; indicating, that only amino acids are interacting with the histidine tag (*Figure 1—figure supplement 4D*). Nevertheless, most of our cell culture experiments were performed with nuclear localized mRuby2, which lacks a C-terminal histidine tag, and these measurements correlated with those using the histidine tagged jAspSnFR3-mRuby3 construct (*Figure 1—figure supplement 1D*).

A recently described and concurrently developed biosensor for aspartate is reported to be adversely affected by temperatures higher than 30°C, causing lower maximum Δ*F/F* (*Hellweg et al., 2023*). Our aspartate sensor appears unaffected by temperature up to 37°C, with the same maximum Δ*F/F* at 37°C as compared to 30°C (*Figure 1—figure supplement 2C*). Like all cpGFP-based sensors, it is sensitive to pH but changes in fluorescence due to aspartate far exceed what one might expect from changes in fluorescence due to physiologically attainable changes in intracellular pH (*Figure 1—figure supplement 2D*). To determine whether it had the potential to serve as an aspartate biosensor in mammalian cells, we expressed jAspSnFR3 in H1299 cells along with nuclear RFP. Expression of jAspSnFR3 had no obvious toxic effects and H1299 jAspSnFR3 cells had visible fluorescence in the green channel (*Figure 1C*). As it is the primary substrate for aspartate production, glutamine removal is expected to deplete aspartate levels. Indeed, we found that 24 hr of glutamine withdrawal abolished GFP signal while leaving RFP unchanged. These findings therefore supported the further testing of jAspSnFR3 as a method to quantify aspartate levels over time in live mammalian cells.

## jAspSnFR3 reveals the temporal dynamics of aspartate limitation

Having shown that jAspSnFR3-mRuby3 protein can measure the concentration of aspartate in vitro, we wanted to test the usefulness of the sensor in cells. To that end, we generated stable cell lines with constitutive expression of jAspSnFR3-mRuby3 or jAspSnFR3 and nuclear localized RFP, and generated single-cell clones from each to yield cell lines with uniform expression. In each case, we then normalized GFP sensor signal to RFP signal to control for expression differences within and across cell lines. We also noted that normalization with nuclear RFP and RFP fusion were highly correlated, enabling jAspSnFR3 sensor applications where nuclear RFP labeling is desirable for example for counting cells at multiple timepoints using live cell imaging (*Figure 2—figure supplement 1D*). An important motivation for using a biosensor for tracking aspartate changes is to enable temporal measurements on the same subset of live cells, therefore we used an Incucyte S3 which performs live cell imaging under native cell line growth conditions.

Cellular aspartate levels depend on the availability of metabolic precursors and the activity of several metabolic processes. One such process is the generation of a sufficiently large intracellular $NAD^+$/NADH ratio to drive aspartate precursor synthesis, a process normally maintained through mitochondrial respiration or, in its absence, by treatment with exogenous electron acceptors like pyruvate (*Figure 2A*). Genetic alterations and pharmacological treatments that disrupt mitochondrial respiration can decrease $NAD^+$/NADH and aspartate levels, both of which can be partially restored by supplementation with pyruvate (*Sullivan et al., 2015*; *Birsoy et al., 2015*). We thus tested the ability of jAspSnFR3 to quantify depletion of intracellular aspartate abundance upon treatment with the mitochondrial complex I inhibitor rotenone and the partial rescue of aspartate by supplementing cells with pyruvate. Titrating rotenone in H1299 cells, we observed a dose-dependent decrease in sensor fluorescence with increased rotenone, corresponding to the expected decrease in aspartate synthesis capacity, and a partial restoration of fluorescence in cells co-treated with pyruvate (*Figure 2B*). This observation was extended to different cell lines with different rotenone sensitivities, corroborating the observation of decreased sensor fluorescence upon rotenone treatment and rescue by pyruvate supplementation (*Figure 2—figure supplement 1A, B, E*). Furthermore, we tested and confirmed the ability of pyruvate to rescue aspartate levels after an initial depletion phase with rotenone, thus confirming that the sensor is also able to measure the reverse kinetics of aspartate restoration (*Figure 2—figure supplement 1C*).

We next evaluated the ability of the sensor to measure changes in aspartate without requiring treatment with a mitochondrial inhibitor. To this aim, we used CRISPR/Cas9 to generate an H1299 cell line with a double knockout (DKO) of the genes glutamic oxalacetic transaminases 1 and 2 (GOT1/2 DKO), which renders cells unable to synthesize aspartate and therefore dependent on aspartate uptake from the media (*Garcia-Bermudez et al., 2022*). Using these H1299 GOT1/2 DKO cells, we titrated media aspartate and observed that sensor fluorescence decreased upon aspartate withdrawal, approaching a steady state after approximately 11 hr, that corresponded to the aspartate availability in the media (*Figure 2C*). We note that 10 mM media aspartate, a higher concentration than any other amino acid in media, is still unable to rescue sensor signal significantly above aspartate depleted media. This confirm previous observations that aspartate has poor cell permeability and often requires concentrations of 20 mM or more to robustly contribute to intracellular aspartate pools (*Sullivan et al., 2018*).

Finally, we evaluated the ability of the sensor to measure rapid decreases/increases in aspartate levels. In most cancer cell lines, intracellular aspartate is derived primarily from glutamine oxidation, thus making glutamine depletion and restoration an entry point for affecting aspartate levels. Glutamine withdrawal from H1299 cells showed a rapid depletion in sensor fluorescence with a low-level steady state reached after approximately 12–16 hr (*Figure 2E*). In the same cells, glutamine repletion, after different times of glutamine depletion, caused an initial rapid increase in sensor fluorescence followed by a slower adjustment phase before reaching baseline. Particularly interesting is the sensor readings for cells repleted with glutamine 6 hr after withdrawal, showing a rapid increase followed by a decrease, before a slow increase to baseline. We concurrently measured the cell count over time by counting RFP fluorescent nuclei and the cell confluency (*Figure 2—figure supplement 2A, B*). These data confirm that cells remain viable 12 hr after glutamine withdrawal and that proliferation rates are first fully recovered when aspartate levels reach baseline. Interestingly, the proliferation rate undergoes a similar multi-phase recovery as the aspartate sensor signal. We suspect these phases to be driven by low aspartate causing S-phase arrest during glutamine deprivation (*Patel et al., 2016*), and although more evidence is necessary to prove this suspicion, our data show that an aspartate sensor is an excellent tool toward this end.

## Metformin has slower inhibitor kinetics compared to rotenone

Metformin is a commonly used diabetes treatment that has been shown to act as a mitochondrial complex I inhibitor (*Owen et al., 2000*; *El-Mir et al., 2000*; *Andrzejewski et al., 2014*; *Wheaton et al., 2014*) and can decrease intracellular aspartate levels in a dose responsive way (*Gui et al., 2016*). Whereas rotenone is a lipophilic molecule that can cross the cell membrane and act rapidly, metformin is hydrophilic and poorly permeable to most cells, resulting in comparatively delayed kinetics for metformin to decrease mitochondrial respiration in intact cells. As rotenone and metformin are often used interchangeably as complex I inhibitors, we wondered whether they have an equivalent temporal effect on aspartate or if the delayed effects of metformin on mitochondrial inhibition would similarly

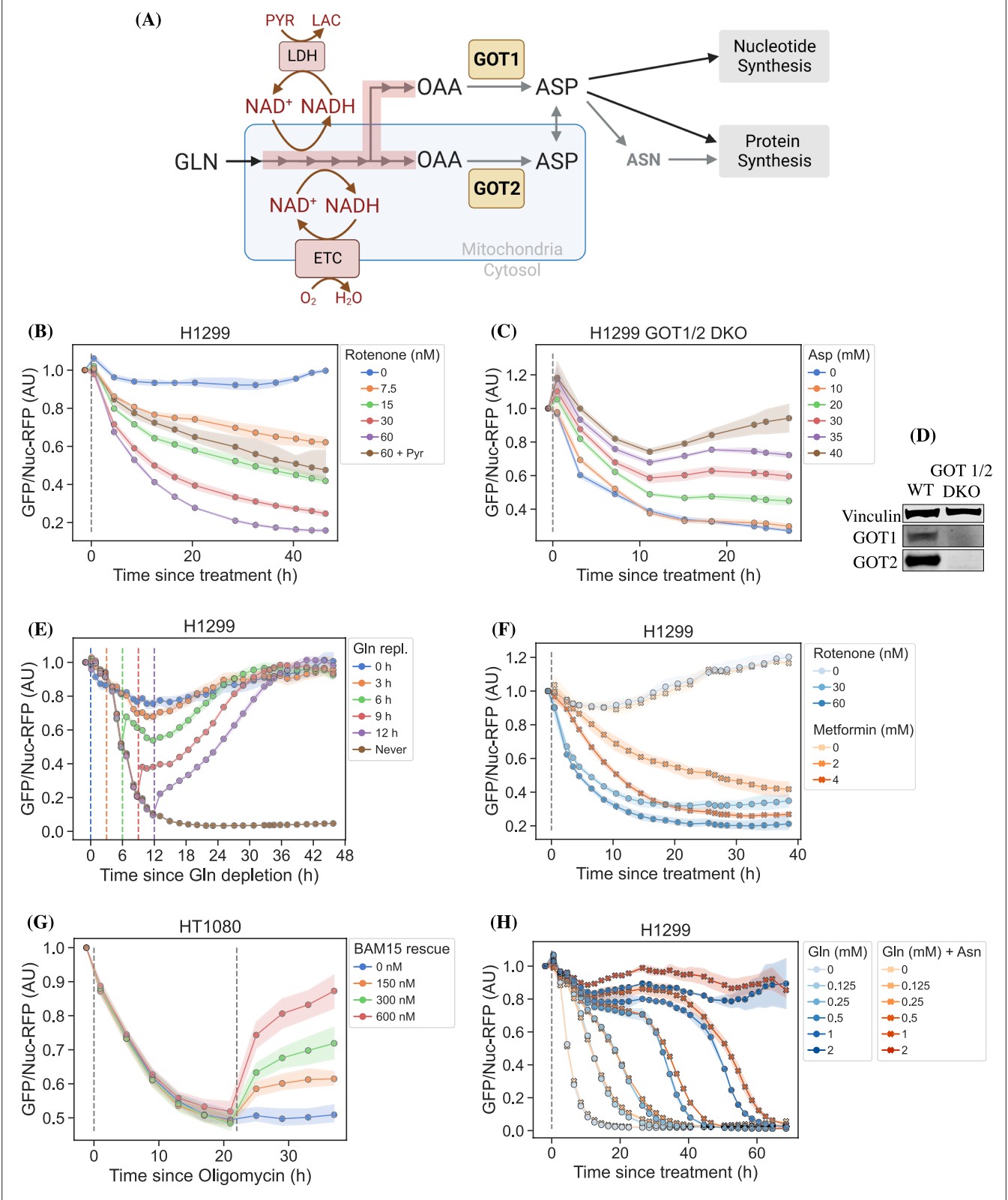

**Figure 2.** jAspSnFR3 resolves temporal aspartate changes in live cells. (**A**) Overview of aspartate metabolism, highlighting how glutamine depletion, mitochondrial inhibition, GOT1/2 knockout and pyruvate/asparagine supplementation can impact aspartate levels. (**B**) H1299 cells treated with a rotenone titration and rescued by co-treatment with pyruvate. (**C**) H1299 GOT1/2 double knockout cells grown in media with 40 mM aspartate, washed thrice in media without aspartate and then changed into media with a titration of aspartate. (**D**) Western blot verification of H1299 GOT1/2 double

*Figure 2 continued on next page*

*Figure 2 continued*

knockout. (**E**) H1299 cells changed into media without glutamine and then glutamine repleted (repl.) at the different timepoints. (**F**) H1299 cells treated with either rotenone or metformin to compare inhibitor kinetics. (**G**) HT1080 cells treated with oligomycin A and after 22 hr rescued with a titration of mitochondrial uncoupler BAM15. (**H**) H1299 cells changed into media with a titration of glutamine with or without 1 mM asparagine. For (**B**), (**C**), (**E**–**H**), sensor signal over time shown as RFP normalized jAspSnFR3 signal following various perturbations of live cells. All experiments shown are normalized to a pre-treatment scan and then treated with the specified drug or amino acid. Gray dashed lines indicate the time of treatment. Colored dashed lines indicate the time of treatment for multiple treatments in the same plot. Markers indicate the average using available well replicates and are superimposed on a bootstrapped 95% confidence interval colored using the same color code as the markers. GLN, glutamine; ETC, electron transport chain; LDH, lactate dehydrogenase; OAA, oxaloacetic acid; ASP, aspartate; ASN, asparagine; AU, arbitrary unit.

The online version of this article includes the following source data and figure supplement(s) for figure 2:

**Source data 1.** Fluorescence measurements for *Figure 2*.

**Source data 2.** Original files for Western blot analysis in *Figure 2D* (anti-Vinculin, anti-GOT1, anti-GOT2).

**Source data 3.** PDF containing *Figure 2D* and original scans of relevant Western blot analysis (anti-Vinculin, anti-GOT1, anti-GOT2) with highlighted bands and sample labels.

**Figure supplement 1.** Rotenone titration in different cell lines.

**Figure supplement 1—source data 1.** Fluorescence measurements for *Figure 2—figure supplement 1*.

**Figure supplement 2.** Plots related to glutamine limitation.

**Figure supplement 2—source data 1.** RFP nuclei counts, confluency measurements, and fluorescence measurements for *Figure 2—figure supplement 2*.

---

delay its effects on aspartate levels. To test this, we treated cells with two doses each of rotenone and metformin with roughly equivalent aspartate lowering effects and followed the sensor signal over time (*Figure 2F*). We observed that the aspartate depleting effects of metformin acted slower than rotenone, with 30 nM rotenone reaching steady state after ~20 hr and 2 mM metformin reaching a similar sensor response after almost 40 hr. These data therefore provide orthogonal confirmation of the differential kinetics of these drugs on cell metabolism and highlight the temporal opportunities enabled by measuring aspartate levels by jAspSnFR3.

## Oligomycin induced decrease in aspartate can be rescued by uncoupling the mitochondrial membrane potential

Oligomycin A binds to and inhibits the proton-coupled rotation of ATP synthase (*Symersky et al., 2012*). This inhibition leads to a slowing of the electron transfer in the mitochondrial electron transport chain by hyperpolarizing the membrane potential (*Brand and Nicholls, 2011*) and it has previously been shown that the cellular effects of oligomycin can be rescued by partial uncoupling of the mitochondrial membrane potential (*Sullivan et al., 2015*; *To et al., 2019*). We used our aspartate sensor to show that oligomycin induces a decrease in aspartate and that the mitochondrial membrane potential specific uncoupler BAM15 (*Kenwood et al., 2014*) can partially restore this aspartate level (*Figure 2G*), thus confirming previous observations and illustrating the versatility of the our sensor.

### Asparagine salvage diverts glutamine consumption

It has previously been reported that asparagine, a product of aspartate metabolism, becomes essential upon glutamine starvation (*Pavlova et al., 2018*; *Zhang et al., 2014*). We hypothesize that asparagine becomes essential in these conditions because glutamine starvation decreases synthesis of aspartate, slowing asparagine production, and because asparagine supplementation spares aspartate consumption, allowing it to be redirected into other essential fates. However, it has been difficult to measure metabolic changes during glutamine limitation because continuous glutamine consumption during the course of the experiment will result in progressive glutamine depletion and further developing metabolic effects. One solution to this problem is to measure the temporal changes in aspartate levels over the course of glutamine starvation and, indeed, we found that full glutamine depletion has a rapid and drastic effect on sensor signal (*Figure 2E, H* and *Figure 2—figure supplement 2C*). Interestingly, aspartate signal did not robustly correlate with the concentration of glutamine in the media in the short term, but instead we found that higher amounts of glutamine in the media delayed the time until aspartate depletion, presumably corresponding to the time at which glutamine is fully depleted and unable to support further aspartate synthesis. Furthermore, we found that adding 1 mM

asparagine delayed the decrease in sensor signal, suggesting that when asparagine can be salvaged from the media it diverts glutamine consumption that would otherwise be purposed for asparagine synthesis via aspartate consumption. We note that, as these data are produced in real time, the method can be used to dynamically find the optimal sampling times to measure and compare intracellular levels of other metabolites relevant to glutamine starvation using MS.

## jAspSnFR3 signal correlates with intracellular aspartate concentration

It is an important requirement for an aspartate sensor that it reflects the intracellular concentration of aspartate over a biologically relevant range for several cell lines. Reference points for the intracellular aspartate concentration can be generated using metabolite extraction and LCMS, but it is important to note that this technique reports the total amount of aspartate summed across all compartments, which can differ in their aspartate concentration (*Chen et al., 2016*). The LCMS-derived concentration also does not reflect protein crowding, aspartate binding to enzymes, or other factors that would affect the free aspartate concentration. Nevertheless, LCMS is the standard approach in studying metabolism and has previously been used to correlate aspartate levels with cell proliferation (*Gui et al., 2016*; *Hart et al., 2023*). Thus, we titrated mitochondrial inhibitors of complex I (rotenone and metformin) and complex III (antimycin A), with or without pyruvate rescue, in three different cell lines and waited

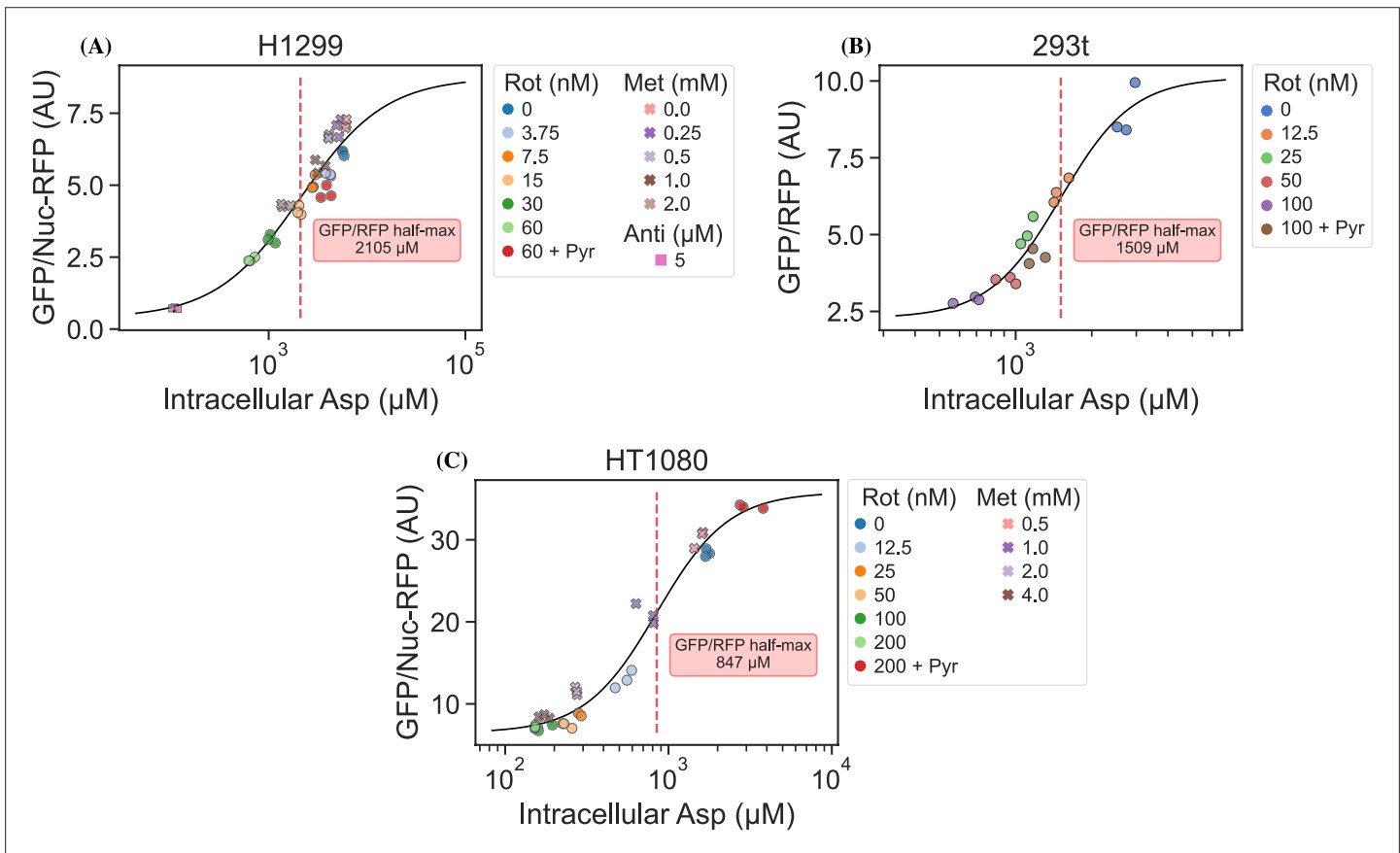

**Figure 3.** jAspSnFR3 signal predicts liquid chromatography–mass spectrometry (LCMS) measured intracellular aspartate concentration. (**A**) Hill equation with top and bottom asymptotes, midpoint and slope as free variables is fitted to the data points and shown by the black line. The intracellular aspartate concentration at the inferred half maximum of RFP normalized jAspSnFR3 signal is reported in the red inserts. (**A**) Rotenone, metformin, and antimycin A titrations in H1299 cells. (**B**) Rotenone titration in HEK293t cells. (**C**) Rotenone and metformin titrations in HT1080 cells. Markers indicate a single well from which both LCMS and jAspSnFR3 data was collected. Replicate wells have identical color and marker shape. AU, arbitrary unit.

The online version of this article includes the following source data and figure supplement(s) for figure 3:

**Source data 1.** Fluorescence measurements and intracellular aspartate measurements for *Figure 3*.

**Figure supplement 1.** jAspSnFR3 signal does not correlate with glutamate concentration.

**Figure supplement 1—source data 1.** Fluorescence measurements and intracellular glutamate measurements for *Figure 3—figure supplement 1*.

24 hr until aspartate had reached near steady-state levels before conducting a final measurement of sensor fluorescence, followed by immediate metabolite extraction and quantitative LCMS measurements of aspartate levels using isotope dilution. We then compared sensor signals to LCMS-derived aspartate concentrations and fitted a Hill curve to infer the intracellular aspartate concentrations at half-maximum sensor signal (*Figure 3*). For all three cell lines, we observe a monotonically increasing relationship between sensor signal and intracellular aspartate concentration, covering around two orders of magnitude. We also observe no relationship between sensor signal and intracellular glutamate levels (*Figure 3—figure supplement 1*). These observations validate the utility of our sensor in a biologically relevant range of aspartate concentrations without interference from glutamate.

Interestingly, the intracellular aspartate concentrations at half-maximum sensor fluorescence are more than 17-fold higher than the aspartate $K_D$ determined by in vitro characterization of the sensor. While the numbers inferred for intracellular aspartate are only point estimates, it is highly unlikely that they are inaccurate to this degree. We speculate that the apparent cytosolic aspartate concentration is likely lower than the total aspartate concentration summed across all compartments. This suggests that binding of aspartate by jAspSnFR3 is in competition with other proteins and highlights that another advantage of using a biosensor is that measurements are made relative to their native environment.

## Discussion

A biosensor for aspartate is an important step toward improved understanding of aspartate metabolism. We have shown that our jAspSnFR3 sensor can resolve temporal changes in intracellular aspartate to answer questions that would be impractical using LCMS. In most studies involving aspartate metabolism, metabolite extraction is performed 6–16 hr after treatment with the implicit assumption that this is enough time to reach metabolic steady state. Using our sensor, we have shown that the time to reach steady state can be much longer and depends on the treatment. Future studies seeking to understand the effects of treatments affecting aspartate levels can therefore use real-time measurements using jAspSnFR3 to determine when cells have reached steady state and then perform metabolite extraction for LCMS.

Recently, a concurrently developed aspartate sensor based on SF-iGluSnFR was reported by another group (*Hellweg et al., 2023*). That aspartate sensor started with the S72A mutation, but then included six additional mutations identified by a combination of targeted screening and deep mutational scanning. Our design differs from the one by *Hellweg et al., 2023* by 25 mutations (*Supplementary file 1*). Notably, we achieved aspartate selectivity by generating only two mutations in the binding pocket of its precursor, and so the majority of these amino acid differences derive from us building off the next-generation precursor for iGluSnFR3, which has increased Δ*F/F* compared to iGluSnFR. Indeed, the sensor described here appears to have a larger signal change both in vitro and in live cells, although head-to-head comparisons have not been performed. Importantly, while the sensor from *Hellweg et al., 2023* is adversely affected by temperature at 37°C, our sensor is not, allowing us to perform cell culture experiments at standard incubation conditions. Another difference is that our jAspSnFR3 sensor has a higher affinity for all three relevant ligands (aspartate, asparagine, and glutamate); however, we found no discernible effect of treatment with 1 mM asparagine on sensor signal in cell culture experiments and found no correlation between intracellular glutamate concentration and sensor fluorescence across treatment conditions (*Figure 3—figure supplement 1*). Collectively, we conclude that the jAspSnFR3 aspartate sensor reported here has biochemical features that makes it ideal for measuring intracellular aspartate levels in live cells.

In summary, we report a novel fluorescence based biosensor that enables dynamic measurements of aspartate. This tool is free of significant interference from relevant metabolites in physiological intracellular systems and can resolve changes in aspartate from diverse treatment conditions in live cells over time. This approach to measuring aspartate will also have advantages compared to LCMS-based metabolomics, including enabling high-throughput experiments to identify variables that affect aspartate levels, such as testing the effects of a drug library on cells in multiwell plates or using fluorescence-activated cell sorting (FACS)-based selection during genetic screens. Another potential use for this sensor would be to dissect compartmentalized metabolism, with mitochondria being a critical target, although incorporating the influence of pH on sensor fluorescence will be an important

consideration in this context. Altogether, adoption of jAspSnFR3 to measure aspartate levels will therefore provide novel opportunities to understand this critical node of cell metabolism.

# Materials and methods

## Key resources table

| Reagent type (species) or resource | Designation | Source or reference | Identifiers | Additional information |
| --- | --- | --- | --- | --- |
| Cell line (human) | H1299 | ATCC | CRL-5803 | |
| Cell line (human) | HEK293T | ATCC | CRL-1573 | |
| Cell line (human) | HT1080 | ATCC | CCL-121 | |
| Other | EF1A-nuclear RFP lentivirus | Cellomics Technology | PLV-10205--50 | Virus |
| Other | U-13C, U-15N-labeled canonical amino acid mix | Cambridge Isotope Laboratories | MSK-CAA-1 | Isotopic standard |
| Antibody | anti-Vinculin, mouse monoclonal | Sigma | SAB4200729 | 1:10,000 dilution |
| Antibody | anti-GOT2, rabbit polyclonal | Proteintech | 14800-1-AP | 1:750 dilution |
| Antibody | anti-GOT1, rabbit monoclonal | Cell Signaling | 34423S | 1:1000 dilution |
| Sequence-based reagent | GOT1_1 | Synthego | CRISPR guide | CAGUCAUCCGUGCGAUAUGC |
| Sequence-based reagent | GOT1_2 | Synthego | CRISPR guide | GCACGGAUGACUGCCAUCCC |
| Sequence-based reagent | GOT1_3 | Synthego | CRISPR guide | CGAUCUUCUCCAUCUGGGAA |
| Sequence-based reagent | GOT2_1 | Synthego | CRISPR guide | UUUCUCAUUUCAGCUCCUGG |
| Sequence-based reagent | GOT2_2 | Synthego | CRISPR guide | CGGACGCUAGGCAGAACGUA |
| Sequence-based reagent | GOT2_3 | Synthego | CRISPR guide | UCCUUCCACUGUUCCGGACG |
| Other | Incucyte S3 | Sartorius | | Instrument |
| Other | Q Exactive HF-X Hybrid Quadrupole | Thermo | | Instrument |
| Sequence-based reagent | S72X | IDT | Primer | CAG CAG TGG AAT ACG GTT TTG SNN GGT AAT CGG AAT CAG TTT TAC |
| Sequence-based reagent | S27X.E25K | IDT | Primer | GTC GTA ATA AGA GAA AGG CAC SNN AGA TTT ACG GTG ACC GAC GAC |
| Chemical compound | Aspartate | Sigma-Aldrich | A6683 | |
| Chemical compound | Glutamate | Sigma-Aldrich | G8415 | |
| Chemical compound | Alanine | Sigma-Aldrich | A7627 | |
| Chemical compound | Cysteine | Sigma-Aldrich | C7352 | |
| Chemical compound | Phenylalanine | Sigma-Aldrich | P2126 | |
| Chemical compound | Glycine | Fisher Bioreagents | BP381 | |
| Chemical compound | Histidine | Sigma-Aldrich | H8125 | |
| Chemical compound | Isoleucine | Sigma-Aldrich | I2752 | |
| Chemical compound | Leucine | Sigma-Aldrich | L8000 | |
| Chemical compound | Lysine | Sigma-Aldrich | L5626 | |
| Chemical compound | Methionine | Sigma-Aldrich | M9625 | |
| Chemical compound | Asparagine | Sigma-Aldrich | A4159 | |
| Chemical compound | Proline | Sigma-Aldrich | P5607 | |
| Chemical compound | Glutamine | Sigma-Aldrich | G3126 | |
| Chemical compound | Arginine | Sigma-Aldrich | A5131 | |
| Chemical compound | Serine | Sigma-Aldrich | S4311 | |
| Chemical compound | Threonine | Sigma-Aldrich | T8625 | |

*Continued on next page*

*Continued*

| Reagent type (species) or resource | Designation | Source or reference | Identifiers | Additional information |
|---|---|---|---|---|
| Chemical compound | Valine | Sigma-Aldrich | V0513 | |
| Chemical compound | Tyrosine | Fluka | 93830 | |
| Chemical compound | Tryptophan | Sigma-Aldrich | T0254 | |
| Chemical compound | Citrate | Thermo Fisher Scientific | A940 | |
| Chemical compound | L-Lactate | Sigma-Aldrich | L7022 | |
| Chemical compound | Pyruvate | Gibco | 11360070 | |
| Chemical compound | Rotenone | Sigma-Aldrich | R9975 | |
| Chemical compound | Metformin | Sigma-Aldrich | 317240 | |
| Chemical compound | Malate | Sigma-Aldrich | 240176 | |
| Chemical compound | Fumarate | Sigma-Aldrich | F1506 | |
| Chemical compound | alpha-Ketoglutarate | Sigma-Aldrich | 75890 | |
| Chemical compound | cis-Aconitic acid | Sigma-Aldrich | A3412 | |
| Chemical compound | Oxaloacetate | Sigma-Aldrich | O4126 | |
| Chemical compound | Succinate | Sigma-Aldrich | S2378 | |
| Chemical compound | GABA | Sigma-Aldrich | A2129 | |

## Sensor engineering and screening

The starting template for jAspSnFR3 was a variant along the path of making iGluSnFR3 (sequence information in *Supplementary file 1*). Site saturation mutagenesis at positions S72 and S27 was achieved by the uracil template method (*Kunkel, 1985*). Mutant libraries (maximum theoretical diversity of 20 each) were transformed into T7 express cells. Individual colonies were picked into a 96-well plate containing auto-induction media (*Studier, 2005*) and shaken at 30°C for 18–24 hr, then harvested by centrifugation. Cell pellets were resuspended in phosphate-buffered saline (PBS) and repelleted by centrifugation five times over, then frozen as pellets overnight. The frozen 96-well plate was thawed by addition of room temperature PBS, agitated by vortexing to resuspend and lyse cells, and then pelleted again. 100 µl clarified lysate was added to each of two black 96-well plates and its fluorescence was measured. Aspartate or glutamate was added (final concentration 100 µM) and fluorescence was measured again. Wells that had a higher $\Delta F/F$ for aspartate than glutamate were isolated, titrated with aspartate and glutamate, and sequenced. After confirming that S72P was the most selective variant for aspartate from a library of S72X, a library of S27X was made in the background of S72P. The selection process was repeated, and S72*P* + S27A was identified as the 'best' aspartate sensor and named jAspSnFR3. mRuby3 was subsequently cloned at the C-terminus and this construct was named jAspSnFR3-mRuby3.

## Protein expression and purification

For large-scale protein expression and purification, jAspSnFR3-mRuby3 was transformed into T7 express cells and a single colony was grown in 300 ml auto-induction media (*Studier, 2005*) at 30°C for ~18 hr. Cells were pelleted by centrifugation at $6000 \times g$, resuspended in PBS and 1 M NaCl and frozen. The resuspended cell pellet was thawed, sonicated on ice (5 s on, 5 s off, 10 min), and centrifuged at $6000 \times g$ to remove cellular debris. The lysate was further clarified by centrifugation at $35,000 \times g$ for 1 hr, and then purified by IMAC on a HisTrap FF column, with a 2 ml/min flow rate and elution from 0 to 200 mM imidazole over 120 ml. Untagged mRuby2 and mRuby3 were partially purified by diethylaminoethyl cellulose (DEAE) anion exchange. Fluorescent fractions were pooled, concentrated by ultrafiltration, and dialyzed in PBS to remove endogenously bound ligands. Protein concentration was determined by alkaline denaturation, and measurement at A447 (Ext. Coeff. 44,000 $M^{-1}$ $cm^{-1}$).

## Sensor biochemical characterization

In vitro fluorescence measurements were performed on a Tecan Spark plate reading fluorimeter at ~28°C, with the exception of the controlled temperature measurement, in which a BioTek Cytation

5 was used. Concentrated jAspSnFR3-mRuby3 protein was diluted to 0.2 µM in PBS for all measurements. Decoy amino acids and pharmacologues were purchased from Sigma-Aldrich and solvated as 100 mM stocks in PBS, with the exception of rotenone, which was resuspended in dimethyl sulfoxide (DMSO). Titrations were performed by making serial dilutions (1:2) of the stock compound into PBS, and adding 10 µl of that to 100 µl of 0.2 µM protein solution. Fluorescence was measured before addition of compound, and $\Delta F/F$ was calculated as $(F(\text{treatment}) - F(\text{initial}))/F(\text{initial})$.

Two-photon cross-sections were collected for 1 µM solutions of protein in PBS with or without 10 mM aspartate, excited by pulses from a mode-locked Ti:Sapphire laser (Chameleon Ultra, Coherent, 80 MHz, 140 fs pulse width, 1 mW power at the focus). Emission was detected by an avalanche photo-diode (PDM Series, Micro Photon Devices) with a 550-nm filter (88 nm bandpass).

Stopped-flow kinetics of binding were determined by mixing equal volumes of 0.2 µM jAspSnFR3 protein (in PBS, pH 7.4) with varying concentrations of aspartate in an Applied Photophysics SX20 stopped-flow fluorimeter with 490 nm LED excitation and 515 nm long pass filter at room temperature (22°C). The increase in fluorescence upon mixing was observed with 1000 data points over the course of 1 s (1 ms per data point) and equilibrium was reached within 100 ms. Each mixing was repeated five times and averaged. Standard deviations are excluded from the plots as the error bars are nearly the same size as the data markers. The first three observations were discarded to remove mixing arte-facts and account for the dead time of the instrument. Data were plotted and time courses were fit to a single rising exponential ($y$-intercept + total rise * $(1 - \exp(-k_{obs}*t))$) using Kaleidagraph (version 5.01). To determine the kinetic rates, $k_{obs}$ was plotted as a function of aspartate concentration and the linear portion of that graph was fitted. The slope, corresponding to the on-rate $k_1$, was found to be 0.8 µM$^{-1}$ s$^{-1}$ and the $y$-intercept, corresponding to the off-rate $k_{-1}$, was found to be 26 s$^{-1}$, resulting in a $K_D$ of 33 µM.

## Cell culture

Cell lines were acquired from ATCC (HEK293T, H1299, HT1080). Cell line identity was verified by STR profiling and cells were regularly tested to be free from mycoplasma contamination (MycoProbe, R&D Systems). Cells were maintained in Dulbecco's modified Eagle's medium (DMEM) (Gibco, 50-003-PB) supplemented with 3.7 g/l sodium bicarbonate (Sigma-Aldrich, S6297), 10% fetal bovine serum (FBS) (Gibco, 26140079) and 1% penicillin–streptomycin solution (Sigma-Aldrich, P4333). Cells were incubated in a humidified incubator at 37°C with 5% $CO_2$.

## Generation of nuclear RFP cell lines

Nuclear RFP cell lines were generated using 1e5 transducing units of EF1A-nuclear RFP lentivirus (Cellomics Technology, PLV-10205-50) by spinfection. Cells were seeded at 50% confluency in 6-well dishes, lentivirus was added to fresh media with 8 µg/µl polybrene, then added to cells and followed by centrifugation (900 × $g$, 90 min, 30°C). Two days after infection, cells were sorted for high RFP expression using FACS. High RFP cells were then expanded and single-cell cloned by limiting dilution, plating 0.5 cells/well on a 96-well plate. Plates were then screened for RFP expression and localization using Incucyte S3 (Sartorius) and a suitable clone chosen, expanded, and used for all subsequent experiments.

## Lentiviral production and stable cell line generation

jAspSnFR3 and jAspSnFR3-mRuby3 were first cloned into entry vector pENTR1A (Fisher, A10462) using NEBuilder HiFI DNA Assembly Cloning Kit (New England BioLabs, E2621). These donor constructs were then used to transfer their insert into destination vectors: pLX304-CMV-Blast (Addgene, 25890), pLenti-CMV-Hygro (w117-1) (Addgene, 17,454 a gift from Eric Campeau & Paul Kaufman), or pLX304-CAG-Blast using LR Clonase II (Fisher, 11791100). pLX304-CAG-Blast was generated in house by swapping the CMV promoter region of pLX304-CMV-Blast with a CAG promoter provided on synthetic DNA (Integrated DNA Technologies). Each plasmid sequence was verified by whole plasmid sequencing (Plasmidsaurus). Lentivirus was generated by co-transfection of HEK293T cells with destination vector plasmid DNA and the packaging plasmids pMDLg/pRRE (Addgene, 12251), pRSV-Rev (Addgene, 12253), and pMD2.G (Addgene, 12259) using FuGENE transfection reagent (Fisher, PRE2693) in DMEM (Fisher, MT10017CV) without FBS or penicillin–streptomycin. The supernatant containing lentiviral particles was filtered through a 0.45-µM membrane (Fisher, 9720514) and

**Table 1.** CRISPR guides.

| Gene | sgRNA sequence (5′–3′) |
| --- | --- |
| GOT1 | CAGUCAUCCGUGCGAUAUGC<br>GCACGGAUGACUGCCAUCCC<br>CGAUCUUCUCCAUCUGGGAA |
| GOT2 | UUUCUCAUUUCAGCUCCUGG<br>CGGACGCUAGGCAGAACGUA<br>UCCUUCCACUGUUCCGGACG |

was supplemented with 8 µg/µl polybrene (Sigma, TR-1003-G) prior to infection. For infection, cells were seeded at 50% confluency in 6-well dishes and centrifuged with lentivirus (900 × *g*, 90 min, 30°C). After 24 hr the media was replaced with fresh media and after 48 hr cells were treated with either 1 µg/ml blasticidin (Fisher, R21001) or 150 µg/ml hygromycin (Sigma-Aldrich, H7772-1G) and maintained in selection media until all uninfected control cells died. After selection, cells were expanded and single-cell cloned by limiting dilution, plating 0.5 cells/well using two to three 96-well plates. These clones were incubated until 10–30% confluency and screened for high GFP and RFP signal using Incucyte S3 (Sartorius). The highest expressing monoclonal cells were selected and further expanded on 6-well plates and again screened for fluorescence using the Incucyte. From this a single clone was chosen, expanded, and used for all subsequent experiments. Different cell lines received different vector-sensor combinations: HEK293T cells were infected with pLX304-CAG-jAspSnFR3-mRuby3 (blasticidin), HT1080 with pLenti-jAspSnFR3-mRuby3 (hygromycin), and HT1080, H1299, and H1299 GOT1/2 DKO cells expressing nuclear RFP were infected with pLenti-jAspSnFR3 (hygromycin).

## Generation of GOT1/2 DKO cells

Protocol and guide RNA generation was identical to that described in *Hart et al., 2023*. Briefly, three chemically synthesized 2′-*O*-methyl 3′phosphorothioate-modified single-guide RNA (sgRNA) sequences targeting GOT1 and GOT2 were purchased (Synthego; *Table 1*). A pool of all six sgRNAs for GOT1 and GOT2 were resuspended in nuclease-free water, combined with SF buffer (Lonza, V4XC-2032), and sNLS-spCas9 (Aldevron, 9212). 200,000 H1299 cells were resuspended in the resulting solution containing ribonucleoprotein complexes (RNPs) and electroporated using a 4D-Nucleofector (Amaxa, Lonza). Nucleofected cells were then expanded and single-cell cloned by limiting dilution by plating 0.5 cells/well in a 96-well plate. Gene knockout was confirmed using Western blots.

## Intracellular jAspSnFR3 measurements

Experiments were conducted in DMEM without pyruvate (Corning 50-013-PB) supplemented with 3.7 g/l sodium bicarbonate 10% dialyzed FBS (Sigma-Aldrich, F0392) and 1% penicillin–streptomycin solution. To start an experiment, cells were trypsinized (Corning, 25,051 CI), resuspended in media, counted using a coulter counter (Beckman Coulter, Multisizer 4) and seeded onto 24-well dishes (Nunc, 142475) with an initial seeding density of 50,000, 70,000, 70,000, or 150,000 cells/well for H1299, H1299 GOT1/2 DKO, HT1080, and HEK293T, respectively. After 24 hr (H1299, HT1080, HEK293T) or 48 hr (H1299 GOT1/2 DKO) incubation, treatment was added and plates moved into an Incucyte S3 (Sartorius) live cell imaging platform inside a humidified incubator at 37°C with 5% $CO_2$. Rotenone (Sigma-Aldrich, R8875), metformin (Sigma-Aldrich, D150959), antimycin A (Sigma-Aldrich, A8674), oligomycin A (Sigma-Aldrich, 495455), and BAM15 (Cayman Chemical, 17811) treatments were spiked-in as 20× solutions in water and the 2 mM pyruvate (Sigma-Aldrich, P8574) was added as 500× stock in water. For treatments with varying media aspartate (Sigma-Aldrich, A7219) or glutamine (Sigma-Aldrich, G5792), wells were washed thrice and filled with media deplete of the given amino acid, then it was added as a spike-in at the specified concentration from a 20× solutions in water. For plates receiving asparagine (Sigma-Aldrich, A7094), this was added to 1 mM from a 20× solution in water, with vehicle wells receiving water. Live cell imaging was performed on the Incucyte S3 using the GFP and RFP channels with default exposure times. Images were processed using the associated Incucyte software to subtract background, define areas of cell confluence and GFP/RFP signal and extract the sum of the fluorescence signal in these areas. The data for the GFP signal, RFP signal, GFP/RFP ratio, and confluence for each well at each timepoint were exported and used for further data

processing using Python code. The jAspSnFR3 signal (GFP channel) was normalized to an RFP signal, either as a stably expressed nuclear localized RFP (Nuc-RFP) or mRuby3 C-term fusion to jAspSnFR3. For some experiments, the Nuc-RFP was used to estimate cell counts per well by counting the number of nuclear foci in each field of view when scanning in the RFP channel. Cell confluency was determined with the phase contrast scans using the associated Incucyte software. For some experiments, a pre-treatment scan was made shortly prior to treatment to normalize the data to this point. For temporal measurements without a pre-treatment scan, the first scan was made 30 min after treatment with subsequent scans indicated on relevant plots. For comparisons of near steady-state measurements of GFP/RFP versus MS-based metabolite measurements, a single scan was made 24 hr after treatment, the plate was then quickly moved to ice and metabolite extraction performed (see below). Another plate was processed in parallel for cell volume determination using a coulter counter and averaging across three replicate wells. The normalized jAspSnFR3 signal as a function of intracellular aspartate concentration, $f(c)$, was fitted by a baseline shifted Hill curve:

$$f(c) = t + \frac{b - t}{1 + (c/m)^s}$$

With $t$, $b$ being the top and bottom of the curve, respectively, describing the upper and lower asymptotes of normalized jAspSnFR3 signal. The curve slope is described by $s$, also known as Hill coefficient, and the midpoint ($m$) describes the intracellular aspartate concentration at half maximum jAspSnFR3 signal. The curve parameters were fitted to the data using the Broyden–Fletcher–Goldfarb–Shanno (BFGS) algorithm with an upper bound constraint on the top of the curve of 1.2 times the maximum observed normalized jAspSnFR3 signal in any of the conditions on the same plot. Note that this curve is not intended to represent a mechanistic model of the binding kinetics, rather the purpose is to infer a reasonable estimate of the intracellular aspartate concentration at half maximum jAspSnFR3 signal.

## Metabolite extraction

For polar metabolite extraction, a plate was moved to ice and the media was thoroughly aspirated. For H1299 and HT1080 cells, wells were washed once with cold saline (Fisher, 23293184). For HEK293T cells, washing was omitted due to weak cell adherence. Then, 1 ml 80% high-performance liquid chromatography (HPLC) grade methanol in HPLC grade water was added, cells were scraped with the back of a P1000 pipet tip and transferred to Eppendorf tubes. Tubes were centrifuged (17,000 × $g$, 15 min, 4°C) and 800 µl of the supernatant containing polar metabolites was transferred to a new centrifuge tube and placed in a centrivap until dry.

## Intracellular amino acid concentration measurements by isotope dilution

Dried samples were reconstituted with 40 µl 80% HPLC grade methanol containing 5 µM U-13C, U-15N-labeled canonical amino acid mix (Cambridge Isotope Laboratories, MSK-CAA-1) and transferred to vials for measurement by LCMS. The peak area for each amino acid was divided by its labeled standard to derive the response ratio. The response ratio was then mapped to a calibration curve to infer the amino acid concentration and finally the intracellular concentration was calculated by correcting for each step introducing a dilution, including the use of the total cell volume. To make the calibration curves a non-labeled amino acid mixture was made from an analytical amino acid standard without glutamine and asparagine (Sigma-Aldrich, A9906-1ML) and added glutamine (Sigma-Aldrich, 76523-100 MG) and asparagine (Sigma-Aldrich, 51363-100 MG) to match the concentration of the other amino acids. Using this mix, three replicates of a 12-point 2-fold dilution series was made with a max concentration of 500 µM and a volume per dilution of 40 µl. These were placed in a centrivap until dry and reconstituted with 40 µl 80% HPLC grade methanol containing 5 µM U-13C, U-15N-labeled canonical amino acid mix (Cambridge Isotope Laboratories, MSK-CAA-1) and transferred to vials for measurement by LCMS. The peak area for each amino acid was divided by its labeled standard to derive the response ratio, then the best fitting calibration curves for each amino acid were chosen among either linear, power, or a second-degree polynomial. Each calibration curve was manually inspected for proper fit and measurements below or above the concentration range of the dilution series were discarded.

## Liquid chromatography–mass spectrometry

Metabolite quantitation was performed using a Q Exactive HF-X Hybrid Quadrupole-Orbitrap Mass Spectrometer equipped with an Ion Max API source and H-ESI II probe, coupled to a Vanquish Flex Binary UHPLC system (Thermo Scientific). Mass calibrations were completed at a minimum of every 5 days in both the positive and negative polarity modes using LTQ Velos ESI Calibration Solution (Pierce). Polar samples were chromatographically separated by injecting a sample volume of 1 μl into a SeQuant ZIC-pHILIC Polymeric column (2.1 × 150 mm 5 mM, EMD Millipore). The flow rate was set to 150 ml/min, autosampler temperature set to 10°C, and column temperature set to 30°C. Mobile Phase A consisted of 20 mM ammonium carbonate and 0.1% (vol/vol) ammonium hydroxide, and Mobile Phase B consisted of 100% acetonitrile. The sample was gradient eluted (%B) from the column as follows: 0–20 min: linear gradient from 85% to 20% B; 20–24 min: hold at 20% B; 24–24.5 min: linear gradient from 20% to 85% B; 24.5 min-end: hold at 85% B until equilibrated with ten column volumes. Mobile Phase was directed into the ion source with the following parameters: sheath gas = 45, auxiliary gas = 15, sweep gas = 2, spray voltage = 2.9 kV in the negative mode or 3.5 kV in the positive mode, capillary temperature = 300°C, RF level = 40%, auxiliary gas heater temperature = 325°C. Mass detection was conducted with a resolution of 240,000 in full scan mode, with an AGC target of 3,000,000 and maximum injection time of 250 ms. Metabolites were detected over a mass range of 70–850 $m/z$. Quantitation of all metabolites was performed using Tracefinder 4.1 (Thermo Scientific) referencing an in-house metabolite standards library using ≤5 ppm mass error.

## Data analysis and plotting

All data processing, curve fitting, plotting, and statistics for experiments involving jAspSnFR3 expressed in cell lines were made using Python code and data available on Github: https://www.github.com/krdav/Aspartate-sensor (copy archived at *Davidsen, 2024*).

## Plasmid availability

Submitted to Addgene under article ID 28238106.

## Acknowledgements

We would like to acknowledge Kaspar Podgorski for pre-publication plasmid access to iGluSnFR3 variants and Ronak Patel for the 2 P spectral analysis. Funding This research was supported by the Proteomics & Metabolomics Shared Resource of the Fred Hutch/University of Washington/Seattle Children's Cancer Consortium (P30 CA015704). LBS acknowledges support from the National Institute of General Medical Sciences (NIGMS; R35GM147118). JSM and TAB are funded by the Howard Hughes Medical Institute.

## Additional information

### Funding

| Funder | Grant reference number | Author |
| --- | --- | --- |
| Howard Hughes Medical Institute | | Jonathan S Marvin<br>Abhi Aggarwal<br>Timothy A Brown |
| National Cancer Institute | P30CA015704 | Lucas B Sullivan |
| National Institute of General Medical Sciences | R35GM147118 | Lucas B Sullivan |

The funders had no role in study design, data collection, and interpretation, or the decision to submit the work for publication.

### Author contributions

Kristian Davidsen, Conceptualization, Data curation, Formal analysis, Validation, Investigation, Visualization, Methodology, Writing - original draft, Writing - review and editing; Jonathan S Marvin,

Conceptualization, Resources, Data curation, Formal analysis, Validation, Investigation, Visualization, Methodology, Writing - original draft, Writing - review and editing; Abhi Aggarwal, Conceptualization, Resources, Methodology; Timothy A Brown, Supervision, Funding acquisition, Writing - review and editing; Lucas B Sullivan, Conceptualization, Supervision, Funding acquisition, Writing - original draft, Project administration, Writing - review and editing

### Author ORCIDs
Kristian Davidsen (iD) http://orcid.org/0000-0002-3821-6902
Jonathan S Marvin (iD) http://orcid.org/0000-0003-2294-4515
Abhi Aggarwal (iD) http://orcid.org/0000-0002-6336-566X
Timothy A Brown (iD) http://orcid.org/0000-0001-8941-7678
Lucas B Sullivan (iD) http://orcid.org/0000-0002-6745-8222

Reviewer #3 (Public Review): https://doi.org/10.7554/eLife.90024.3.sa1
Author Response https://doi.org/10.7554/eLife.90024.3.sa2

## Additional files

### Supplementary files
- MDAR checklist

- Supplementary file 1. Protein engineering sequence information for jAspSnFR3.

- Supplementary file 2. Ion count quantification for liquid chromatography–mass spectrometry (LCMS) metabolomics measurements used to calculate aspartate and glutamate concentrations in *Figure 3* and *Figure 3—figure supplement 1*.

### Data availability
All data generated or analyzed during this study are included in the manuscript and supporting files; source data files have been provided for all figures.

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
