## [Editor Report · eLife assessment]

This **important** study reports jAspSnFR3, a biosensor that enables high spatiotemporal resolution of aspartate levels in living cells. To develop this sensor, the authors used a structurally guided amino acid substitution in a glutamate/aspartate periplasmic binding protein to switch its specificity towards aspartate. The in vitro and in cellulo functional characterization of the biosensor is **convincing**, but evidence of the sensor's effectiveness in detecting small perturbations of aspartate levels and information on its behavior in response to acute aspartate elevations in the cytosol are still lacking.

---

## [Referee Report · Reviewer #3 (Public Review)]

Summary:

In this manuscript, Davidsen and collaborators introduce jAspSnFR3, a new version of aspartate biosensor derived from iGluSnFR3, that allows to monitor in real-time aspartate levels in cultured cells. A selective amino acids substitution was applied in a key region of the template to switch its specificity from glutamate to aspartate. The jAspSnFR3 does not respond to other tested metabolites and performs well, is not toxic for cultured cells, and is not affected by temperature ensuring the possibility of using this tool in tissues physiologically more relevant. The high affinity for aspartate (KD=50 uM) allowed the authors to measure fluctuations of this amino acid in the physiological range. Different strategies were used to bring aspartate to the minimal level. Finally, the authors used jAspSnFR3 to estimate the intracellular aspartate concentration.

Strengths:

One of the highlights of the manuscript was a treatment with asparagine during glutamine starvation. Although didn`t corroborate the essentiality of asparagine in glutamine depletion, the measurement of aspartate during this supplementation is a glimpse of how useful this sensor can be.

Weaknesses:

Although this is a well-performed study, I have some comments for the authors to address:

1-A red tag version of the sensor (jAspSnFR3-mRuby3) was generated for normalization purposes, with this the authors plan to correct GFP signal from expression and movement artifacts. I naturally interpret "movement artifacts" as those generated by variations in cell volume and focal plane during time-lapse experiments. However, it was mentioned that jAspSnFR3-mRuby3 included a histidine tag that may induce a non-specific effect (responses to the treatment with some amino acids). This suggests that a version without the tag needs to be generated and that an alternative design needs to be set for normalization purposes. A nuclear-localized RFP was expressed in a second attempt to incorporate RFP as a normalization signal. Here the cell lines that express both signals (sensor and RFP) were generated by independent lentiviral transductions (insertions). Unless the number of insertions for each construct is known, this approach will not ensure an equimolar expression of both proteins (sensor and RFP). In this scenario is not clear how the nuclear expression of RFP will help the correction by expression or monitor changes in cell volume. The authors may be interested in attempting a bicistronic system to express both the sensor and RFP.

2-The authors were interested in establishing the temporal dynamics of aspartate depletion by genetics and pharmaceutical means. For the inhibition of mitochondrial complex I rotenone and metformin were used. Although the assays are clearly showing aspartate depletion the report of cell viability is missing. Considering that glutamine deprivation induces arrest in cell proliferation, I think will be important to know the conditions of the cell cultures after 60 hours of treatment with such inhibitors.

3-The pH sensitivity was checked in vitro with jAspSnFR3-mRuby3 and the sensor reported suitable for measurements at physiological pH. It would be an opportunity to revisit the analysis for pH sensitivity in cultured cells using an untagged version of jAspSnFR3 coupled, for example, to a sensor for pH.

4-While the authors take an interesting approach to measuring intracellular aspartate concentration, it will be highly desirable if a calibration protocol can be designed for this sensor. Clearly, glutamine depletion grants a minimal ("zero") aspartate concentration. However, having a more dynamic way for calibration will facilitate the introduction of this tool for metabolism studies. This may be achieved by incorporating a cultured cell that already expresses the transporter or by ectopic expression in the cells that have already been used.

---

## [Author Response]

The following is the authors’ response to the original reviews.

**eLife assessment**
This important study reports jAspSnFR3, a biosensor that enables high spatiotemporal resolution of aspartate levels in living cells. To develop this sensor, the authors used a structurally guided amino acid substitution in a glutamate/aspartate periplasmic binding protein to switch its specificity towards aspartate. The in vitro and in cellulo functional characterization of the biosensor is convincing, but evidence of the sensor's effectiveness in detecting small perturbations of aspartate levels and information on its behavior in response to acute aspartate elevations in the cytosol are still lacking.

We thank the reviewers and editors for the detailed assessment of our work and for their constructive feedback. Most comments have now been experimentally addressed in the revised manuscript, which we feel is substantially improved from the initial draft.

**Public Reviews:**

**Reviewer #1 (Public Review):**
In this manuscript, Davidsen and coworkers describe the development of a novel aspartate biosensor jAspSNFR3. This collaborative work supports and complements what was reported in a recent preprint by Hellweg et al., (bioRxiv; doi: 10.1101/2023.05.04.537313). In both studies, the newly engineered aspartate sensor was developed from the same glutamate biosensor previously developed by the authors of this manuscript. This coincidence is not casual but is the result of the need to find tools capable of measuring aspartate levels in vivo. Therefore, it is undoubtedly a relevant and timely work carried out by groups experienced in aspartate metabolism and in the generation of metabolite biosensors.
**Reviewer #2 (Public Review):**
In this work the IGluSnFR3 sensor, recently developed by Marvin et al (2023) is mutated position S72, which was previously reported to switch the specificity from Glu to Asp. They made 3 mutations at this position, selected a S72P mutant, then made a second mutation at S27 to generate an Asp-specific version of the sensor. This was then characterized thoroughly and used on some test experiments, where it was shown to detect and allow visualization of aspartate concentration changes over time. It is an incremental advance on the iGluSnFR3 study, where 2 predictable mutations are used to generate a sensor that works on a close analog of Glu, Asp. It is shown to have utility and will be useful in the field of Asp-mediated biological effects.
**Reviewer #3 (Public Review):**
In this manuscript, Davidsen and collaborators introduce jAspSnFR3, a new version of aspartate biosensor derived from iGluSnFR3, that allows monitoring in real-time aspartate levels in cultured cells. A selective amino acids substitution was applied in a key region of the template to switch its specificity from glutamate to aspartate. The jAspSnFR3 does not respond to other tested metabolites and performs well, is not toxic for cultured cells, and is not affected by temperature ensuring the possibility of using this tool in tissues physiologically more relevant. The high affinity for aspartate (KD=50 uM) allowed the authors to measure fluctuations of this amino acid in the physiological range. Different strategies were used to bring aspartate to the minimal level. Finally, the authors used jAspSnFR3 to estimate the intracellular aspartate concentration. One of the highlights of the manuscript was a treatment with asparagine during glutamine starvation. Although didn't corroborate the essentiality of asparagine in glutamine depletion, the measurement of aspartate during this supplementation is a glimpse of how useful this sensor can be.
**Reviewer #1 (Recommendations For The Authors):**
The authors should evaluate the effectiveness of the sensor in detecting small perturbations of aspartate levels and its behavior in response to acute aspartate elevations in the cytosol. In vivo aspartate determinations were performed exclusively in conditions that cause aspartate depletion. By means the use of mitochondrial respiratory inhibitors or aspartate withdrawal, it was determined the reliability of the sensor performing readings during relatively long periods, until reaching a steady-state of aspartate-depletion 12-60 hours later. Although in Hellweg and coworkers, it has been demonstrated that a related aspartate sensor could detect increases in aspartate in cell overexpressing the aspartate-glutamate GLAST transporter, the differences reported here between both sensors advise testing whether this aspect is also improved, or not, using jAspSNFR3.Similarly, Davidsen et al. did not test if the sensor can be able to detect transient variations in cytosolic aspartate levels. In proliferative cells aspartate synthesis is linked to NAD+ regeneration by ETC (Sullivan et al., 2015, Cell), indeed the authors deplete aspartate using CI or CIII inhibitors but do not analyze if those are recovered, and increased, after its removal. Furthermore, the sequential addition of oligomycin and uncouplers could generate measurable fluctuations of aspartate in the cytosol.

We agree with the reviewer that only including situations of aspartate depletion in our cell culture experiments provided an incomplete evaluation of the utility of this biosensor. In the revised manuscript we provide three additional experiments using secondary treatments that restore aspartate synthesis to conditions that initially caused aspartate depletion. First, we conducted experiments where cells expressing jAspSnFR3/NucRFP were changed into media without glutamine, inducing aspartate depletion, with glutamine being replenished at various time points to observe if GFP/RFP measurements recover. As expected, glutamine withdrawal caused a decay in the GFP/RFP signal and we found that restoring glutamine caused a subsequent restoration of the GFP/RFP signal at all time points, with each fully recovering the GFP/RFP signal over time (Revised Manuscript Figure 2E). Next, we conducted the experiment suggested by the reviewer, testing whether the published finding, that oligomycin induced aspartate limitation can be remedied by co-treatment with electron transport chain uncouplers, could be visualized using jAspSnFR3 measurements of GFP/RFP. Indeed, after 24 hours of oligomycin induced aspartate depletion, treatment with the ETC uncoupler BAM15 dose dependently restored GFP/RFP signal (Revised Manuscript Figure 2G). Finally, we also measured whether the ability of pyruvate to mitigate the decrease in aspartate upon co-treated with rotenone (Figure 2B) could also be detected in a sequential treatment protocol after aspartate depletion. Indeed, after 24 hours of aspartate depletion by rotenone treatment, the GFP/RFP signal was rapidly restored by additional treatment with pyruvate (Revised Manuscript Figure 2, figure supplement 1C). Collectively, these results provide support for the utility of jAspSnFR3 to measure transient changes in aspartate levels in diverse metabolic situations, including conditions that restore aspartate to cells that had been experiencing aspartate depletion.

**Reviewer #2 (Recommendations For The Authors):**
Weaknesses: Sensor basically identical to iGluSnFR3, but nevertheless useful and specific. The results support the conclusions, and the paper is very straightforward. I think the work will be useful to people working on the effects of free aspartate in biology and given it is basically iGluSnFR3, which is widely used, should be very reproducible and reliable.

We appreciate the reviewer’s comment that sensor is useful for specific detection of aspartate. We agree that the advance of the paper is primarily in demonstrating its utility to measure aspartate, rather than any fundamental innovation on the biosensor approach. We hope the fact that jAspSnFR3 derives from a well validated biosensor (iGluSnFR3) will support its adoption.

**Reviewer #3 (Recommendations For The Authors):**
Although this is a well-performed study, I have some comments for the authors to address:1. A red tag version of the sensor (jAspSnFR3-mRuby3) was generated for normalization purposes, with this the authors plan to correct GFP signal from expression and movement artifacts. I naturally interpret "movement artifacts" as those generated by variations in cell volume and focal plane during time-lapse experiments. However, it was mentioned that jAspSnFR3-mRuby3 included a histidine tag that may induce a non-specific effect (responses to the treatment with some amino acids). This suggests that a version without the tag needs to be generated and that an alternative design needs to be set for normalization purposes. A nuclear-localized RFP was expressed in a second attempt to incorporate RFP as a normalization signal. Here the cell lines that express both signals (sensor and RFP) were generated by independent lentiviral transductions (insertions). Unless the number of insertions for each construct is known, this approach will not ensure an equimolar expression of both proteins (sensor and RFP). In this scenario is not clear how the nuclear expression of RFP will help the correction by expression or monitor changes in cell volume. The authors may be interested in attempting a bicistronic system to express both the sensor and RFP.

The reviewer noted several potential issues concerning the use of RFP for normalization, which will be separated into sections below:

Movement artifacts:

We are glad the reviewer raised this issue since we see how it was confusingly worded. We have deleted the text “and movement artefacts” from the sentence.

His-tag and non-specific responses to some amino acids:

We also found it concerning that non-specific responses to amino acids could potentially contribute to our RFP normalization signal, and so we conducted additional experiments to address whether this was likely to be an issue in intracellular measurements. We first tested whether the non-specific signal was related to the histidine tag, or was intrinsic to the mRuby3 protein itself, by comparing the fluorescence response to a titration of histidine (which showed the largest effect of red fluorescence), aspartate, and GABA (structurally related to glutamate and aspartate, but lacking a carboxylate group) across a group of mRuby containing variants, with or without histidine tags. We replicated the non-specific signal originally observed in jAspSnFR3-mRuby3-His and found that another biosensor with a histidine tagged on the C terminus of mRuby3 had a similar response (iGlucoSnFR2.mRuby3-His), as did mRuby3-His alone, indicating that the aspect of being fused with jAspSnFR3 or another binding protein was not required for this effect. Additionally, we also compared the fluorescence response of lysates expressing mRuby2 and mRuby3 without histidine tags and found that the non-specific signal was essentially absent (Revised Manuscript Figure 1, figure supplement 4B-D). Collectively. These data support our original hypothesis that the histidine tag was responsible for the non-specific signal, alleviating concerns about more substantial protein design issues or with using nuc-RFP for normalization. Since we also found that measuring aspartate signal using GFP/RFP ratios from cells with linked the jAspSnFR3-Ruby3-His agreed with measurements from cells separately expressing jAspSnFR3 and nucRFP (without a His tag), and the amino acid concentrations needed to significantly alter His tagged Ruby3 signal are above those typically found in cells, we conclude that this is unlikely to be a significant factor in cells. Nonetheless, we have added all the relevant data to the manuscript to allow readers to make their own decision about which construct would be best for their purposes.

Original text:

"Surprisingly, the mRuby3 component responds to some amino acids at high millimolar concentrations, indicating a non-specific effect, potentially interactions with the C-terminal histidine tag (Figure 1—figure Supplement 2, panel B). Notably, this increase in fluorescence is still an order of magnitude lower than the green fluorescence response and it occurs at amino acid concentrations that are unlikely to be achieved in most cell types."

Revised text:

"Surprisingly, the mRuby3 fluorescence of affinity-purified jAspSnFR3.mRuby3 responds to some amino acids at high millimolar concentrations, indicating a non-specific effect (Figure 1—figure Supplement 4, panel A). This was determined to be due to an unexpected interaction with the C-terminal histidine tag and could be reproduced with other proteins containing mRuby3 and purified via the same C-terminal histidine tag (Figure 1—figure Supplement 4, panel B and C). Interestingly, a structurally related, non-amino acid compound, GABA, does not elicit a change in red fluorescence; indicating, that only amino acids are interacting with the histidine tag (Figure 1—figure Supplement 4, panel D). Nevertheless, most of our cell culture experiments were performed with nuclear localized mRuby2, which lacks a C-terminal histidine tag, and these measurements correlated with those using the histidine tagged jAspSnFR3-mRuby3 construct (Figure 1—figure Supplement 1 panel D)."

Lentiviral transductions

We agree that splitting the two fluorescent proteins across two expression constructs and infections effectively guarantees that there will not be equimolar expression of jAspSnFR3 and RFP, however we do not think equimolar expression is necessary in this context. The primary goal of RFP measurements in these experiments (and in experiments using the jAspSnFR3-mRuby3 fused construct) is to control for global alterations in protein expression that might confound the interpretation that a change in GFP fluorescence corresponds to a change in aspartate levels. While a bicistronic system is arguably a better approach to improve the similarity of expression of jAspSnFR3 and nuc-RFP in a cell, we only require that the cells have consistent expression of both proteins across all cells in the population, not that the expression of one necessarily be a similar molarity to the other. We accomplish consistent expression of proteins by single cell cloning after expression of jAspSnFR3 and nucRFP (or jAspSnFR3-mRuby3), and screening for clones that have high enough expression of both proteins such that they are well detected by standard Incucyte conditions. Given that our data do not identify an obvious downside to separate expression of jASPSnFR3 and nuc-RFP compared to the fused jAspSnFR3-mRuby3 construct (where the fluorescent proteins are truly equimolar) (Figure 2, Figure Supplement 1C), we elected to prioritize the separate jAspSnFR3 and nuc-RFP combination, which provides additional opportunities to measure cell number in the same experiment (see below).

2. The authors were interested in establishing the temporal dynamics of aspartate depletion by genetics and pharmaceutical means. For the inhibition of mitochondrial complex I rotenone and metformin were used. Although the assays are clearly showing aspartate depletion the report of cell viability is missing. Considering that glutamine deprivation induces arrest in cell proliferation, I think will be important to know the conditions of the cell cultures after 60 hours of treatment with such inhibitors.

We agree that ensuring that cells are still viable in conditions where aspartate is depleted, as determined by GFP/RFP in jAspSnFR3 expressing cells, is an important goal. To this end, we added a new experiment investigating the restoration of glutamine on the GFP/RFP signal at different time points after glutamine depletion (Revised Manuscript Figure 2E, see response to reviewer 1). One advantage of using the nuclear RFP as a normalization marker is that it also enables measurements of nuclei counts, a surrogate measurement for cell number. In the same glutamine depletion experiment we therefore measured cell counts using nuclear RFP incidences and confluency as measurements of cell proliferation/growth. In both cases, the arrest in cell proliferation upon glutamine withdrawal was obvious, as was the restoration of cell proliferation following glutamine replenishment, with the amount of growth delay corresponding to the length of glutamine withdrawal (Revised Manuscript Figure 2, Figure Supplement 2A-B). Nonetheless, there was no obvious lasting defects in restarting cell proliferation even after 12 hours of glutamine withdrawal, indicating that cell viability is preserved. In the case of mitochondrial inhibitors, we also observe even that after 24 hours of treatment with oligomycin or rotenone, restoration of aspartate synthesis from BAM15 or pyruvate, respectively, can also restore GFP/RFP signal, supporting the conclusion that cellular metabolism is still active in these conditions (Revised Manuscript Figure 2G; Revised Manuscript Figure 2, figure supplement 1C).

3. The pH sensitivity was checked in vitro with jAspSnFR3-mRuby3 and the sensor reported suitable for measurements at physiological pH. It would be an opportunity to revisit the analysis for pH sensitivity in cultured cells using an untagged version of jAspSnFR3 coupled, for example, to a sensor for pH.

We thank the reviewer for the suggestion and agree that pH effects on sensor signal could be a confounding factor in some conditions. Unfortunately, measuring intracellular pH is not trivial and using multiple fluorescent sensors that change simultaneously would be complex to interpret, particularly in the absence of controls to unambiguously control intracellular pH and aspartate concentrations. Thus, we believe that proper investigation of the variable of pH is beyond the scope of this study. Nonetheless, we agree that measuring the contribution of pH to sensor signal is an important goal for future work, particularly if deploying it in conditions likely to cause substantial pH differences, such as comparing compartmentalized signal of jAspSnFR3 in the cytosol and mitochondria. We have added the following italicized text to the conclusions section to underscore this point:

“Another potential use for this sensor would be to dissect compartmentalized metabolism, with mitochondria being a critical target, although incorporating the influence of pH on sensor fluorescence will be an important consideration in this context.”

4. While the authors take an interesting approach to measuring intracellular aspartate concentration, it will be highly desirable if a calibration protocol can be designed for this sensor. Clearly, glutamine depletion grants a minimal ("zero") aspartate concentration. However, having a more dynamic way for calibration will facilitate the introduction of this tool for metabolism studies. This may be achieved by incorporating a cultured cell that already expresses the transporter or by ectopic expression in the cells that have already been used.

We appreciate the suggestion and would similarly desire a calibration protocol to serve as a quantitative readout of aspartate levels from fluorescence signal, if possible. While we do calibrate jAspSnFR3 fluorescence in purified settings, conducting an analogous experiment intracellularly is currently difficult, if not impossible. While we have several methods to constrain the production rate of aspartate (glutamine withdrawal, mitochondrial inhibitors, and genetic knockouts of GOT1 and GOT2), we cannot prevent cells from decreasing aspartate consumption and so cannot get a true intracellular zero to aid in calibration. Additionally, the impermeability of aspartate to cell membranes makes it challenging to specifically control intracellular concentrations using environmental aspartate, and the best-known aspartate transporter (SLC1A3) is concentrative and so has the reciprocal problem. Considering these issues, we are wary of implying to readers that any specific fluorescence measurement can be used to directly interpret aspartate concentration given the many variables that can impact its signal, both related to the biosensor system itself (expression of jAspSnFR3, expression of Nuc-RFP, sensitivity and settings of the fluorescence detector) and based on cell intrinsic variability (differences in basal ASP levels, different sensitivity to treatments, influence of pH, etc.). We maintain that jAspSnFR3 has utility to measure relative changes in aspartate within a cell line across treatment conditions and over time, but absolute quantitation of aspartate still will require complementary approaches, like mass spectrometry, enzymatic assays, or NMR.

5. jAspSnFR3 seems to have the potential to be incorporated easily for several research groups as a main tool. In general, a minor correction to replace F/F with ΔF/F in the text.

Thank you for catching this error, the text has been edited accordingly.